# Levocarnitine Supplementation Suppresses Lenvatinib-Related Sarcopenia in Hepatocellular Carcinoma Patients: Results of a Propensity Score Analysis

**DOI:** 10.3390/nu13124428

**Published:** 2021-12-10

**Authors:** Hironao Okubo, Hitoshi Ando, Eisuke Nakadera, Kenichi Ikejima, Shuichiro Shiina, Akihito Nagahara

**Affiliations:** 1Department of Gastroenterology, Juntendo University Nerima Hospital, Tokyo 177-8521, Japan; enakader@juntendo.ac.jp; 2Department of Cellular and Molecular Function Analysis, Kanazawa University Graduate School of Medical Sciences, Kanazawa 920-8640, Japan; h-ando@med.kanazawa-u.ac.jp; 3Department of Gastroenterology, Juntendo University School of Medicine, Tokyo 113-0034, Japan; ikejima@juntendo.ac.jp (K.I.); sshiina@juntendo.ac.jp (S.S.); nagahara@juntendo.ac.jp (A.N.)

**Keywords:** lenvatinib, sarcopenia, carnitine, hepatocellular carcinoma

## Abstract

This study investigated the inhibitory effect of levocarnitine supplementation on sarcopenia progression in hepatocellular carcinoma (HCC) patients treated with lenvatinib. We evaluated the skeletal muscle index (SMI). After propensity score matching for age, sex, modified albumin-bilirubin grade, baseline presence of sarcopenia, and branched-chain amino acid administration, we selected 17 patients who received levocarnitine supplementation after starting lenvatinib therapy and 17 propensity-score-matched patients who did not receive levocarnitine. Sarcopenia was present in 76% of the patients at baseline. Changes in baseline SMI at 6 and 12 weeks of treatment were significantly suppressed in the group with levocarnitine supplementation compared with those without (*p* = 0.009 and *p* = 0.018, respectively). While there were no significant differences in serum free carnitine levels in cases without levocarnitine supplementation between baseline and after 6 weeks of treatment (*p* = 0.193), free carnitine levels were significantly higher after 6 weeks of treatment compared with baseline in cases with levocarnitine supplementation (*p* < 0.001). Baseline SMI and changes in baseline SMI after 6 weeks of treatment were significantly correlated with free carnitine levels (r = 0.359, *p* = 0.037; and r = 0.345, *p* = 0.045, respectively). Levocarnitine supplementation can suppress sarcopenia progression during lenvatinib therapy.

## 1. Introduction

Hepatocellular carcinoma (HCC) is one of the most common causes of cancer-related death in Asia [1]. Systemic therapy using tyrosine kinase inhibitors (TKIs) is an effective treatment strategy for unresectable HCC [1,2]. Lenvatinib and sorafenib has been widely available worldwide including in Japan for unresectable HCC. Lenvatinib is an oral TKI that inhibits vascular endothelial growth factor receptor 1–3, fibroblast growth factor receptor 1–4, RET, and platelet-derived growth factor receptor ɑ [3], and has been shown to have a high therapeutic effect in clinical practice [4,5].

Sarcopenia is characterized as the loss of muscle mass and strength [6]. In cancer patients, including those with HCC, targeted therapies, such as TKIs, can exacerbate muscle wasting and loss of skeletal muscle mass, which is significantly associated with low tolerability of chemotherapy and poor survival [7]. Moreover, the presence of sarcopenia is a good indicator of nutrition in cirrhotic patients and is closely related to prognosis in HCC patients [6]. The presence of sarcopenia has also been reported to be associated with increased toxicity and prognosis in HCC patients treated with sorafenib [8,9,10,11]. Additionally, a recent report indicated that loss of skeletal muscle mass in HCC patients treated with lenvatinib influences overall survival and the time to treatment failure [12]. Additionally, a recent study found that eight patients treated with lenvatinib had a significant decrease in baseline skeletal muscle mass at 1–3 months after starting treatment [13]. Therefore, an interventional strategy for sarcopenia progression may be recommended for HCC patients to maintain muscle volume during TKI treatment [14].

Carnitine (3-hydroxy-4-N-trimethylammoniobutanoate) plays a pivotal role in the transport of long-chain fatty acids into the mitochondria for subsequent β-oxidation [15], and >95% of total body carnitine is thought to be stored in skeletal muscle [16,17]. Carnitine supplementation could be an important strategy for improving the quality of life of cirrhotic patients, such as those with sarcopenia, muscle cramps, and hepatic encephalopathy [18]. Nevertheless, few studies have reported on interventions for sarcopenia progression in HCC patients undergoing TKI treatment; specifically, no studies to our knowledge have reported on such interventions in HCC patients treated with lenvatinib. Therefore, we aimed to investigate the inhibitory effect of levocarnitine supplementation on the progression and/or development of sarcopenia in HCC patients undergoing lenvatinib therapy.

## 2. Materials and Methods

### 2.1. Patients

In total, 78 Japanese patients with unresectable HCC, who underwent lenvatinib treatment between July 2018 and July 2021 at Juntendo University Nerima Hospital, were evaluated for inclusion in this study. This retrospective study was approved by the Ethical Committee of Juntendo University Nerima Hospital (approval number: 2020071) and was conducted in accordance with the ethical standards of the 1964 Declaration of Helsinki and its later amendments. Informed consent was obtained from all patients involved in the study prior to the initiation of lenvatinib therapy. Patients took oral lenvatinib (LENVIMA; Eisai Co., Ltd., Tokyo, Japan) 12 mg once daily (for bodyweight ≥60 kg) or 8 mg once daily (for bodyweight <60 kg) as an initial dose. Assessments of skeletal muscle mass were performed using computed tomography (CT) to evaluate therapeutic responses; imaging was performed 6 (±1) and 12 (±1) weeks after starting treatment. Because our recent clinical study indicated that lenvatinib therapy for HCC provoked lenvatinib-related carnitine insufficiency during treatment, patients treated with lenvatinib began receiving an oral levocarnitine supplementation of 1500 mg per day in March 2020 [19].

The flowchart of this study is shown in Figure 1. Among the 78 patients, 18 were excluded due to discontinuation of lenvatinib <6 weeks after the first dose. Among the remaining 60 patients, 11 were excluded due to the following criteria: (i) previous treatment with molecular-targeted agents; (ii) initial CT scans were not performed within 2 weeks of beginning lenvatinib therapy; and (iii) subsequent CT scans were not performed within 6 and 12 weeks after starting lenvatinib therapy. 

Ultimately, we selected 22 patients who received oral levocarnitine supplementation and 27 patients who did not receive the supplementation after starting lenvatinib therapy. No patients received levocarnitine supplementation before lenvatinib therapy. Because there was a lack of randomization, differences in baseline parameters between the two groups may have affected the outcomes; therefore, we used propensity score matching (PSM) to estimate clinical outcomes. The patients with levocarnitine supplementation (n = 22) and controls (n = 27) were matched using PSM for age, sex, modified albumin-bilirubin (mALBI) grade, baseline presence or absence of sarcopenia, and branched-chain amino acid (BCAA) administration. After PSM, the two groups were matched as shown in Table 1. The PSM derivation was checked by the Hosmer–Lemeshow test (*p* = 0.189).

### 2.2. Imaging Analysis of Skeletal Muscle Mass

Baseline CT images taken for tumor staging within 2 weeks prior to initiating lenvatinib treatment were used to examine the skeletal muscle index (SMI). A transverse CT image at the third lumbar vertebra (L3) was selected from each scan. Muscles at this level include psoas, erector spinae, quadratus lumborum, transversus abdominis, external and internal obliques, and the rectus abdominis. All images were analyzed with Slice-O-Matic version 5.0 software (Tomovision, Montreal, QC, Canada), which enables specific tissue selection using previously determined Hounsfield units (HU). Skeletal muscle mass was selected and quantified by thresholds between −29 and 150 HU. To suppress inter-observer measuring error, we used AVACS, an algorithm that automatically segments skeletal muscle [20]. Criteria for the sarcopenia group were evaluated by the working group for creating sarcopenia assessment criteria for the Japan Society of Hepatology. The cutoff values for the sarcopenia group were <38 cm^2^/m^2^ for women and <42 cm^2^/m^2^ for men [6]. CT scans were performed 6 (±1) and 12 (±2) weeks after starting lenvatinib therapy.

### 2.3. Measurements of Serum Ammonia and Carnitine Content

Fasting morning blood samples were collected from patients before starting lenvatinib therapy, and at 2, 4, 6, and 12 weeks after the initiation of lenvatinib therapy. Blood samples were centrifuged within 30 min after collection at 3000 rpm for 5 min. Serum ammonia levels were analyzed at our central laboratory by an enzyme method using an autoanalyzer (JCA-BM6070 series; JEOL, Tokyo, Japan). Serum total and free carnitine concentrations were examined at a clinical laboratory of SRL (Tokyo, Japan) by an enzyme cycling method using an autoanalyzer (JCA-BM8040 series; JEOL, Tokyo, Japan). The value of acyl-carnitine was calculated by the difference between total carnitine and free carnitine [21].

### 2.4. Statistical Analysis

Continuous variables are expressed as median values (range) and were analyzed using the Mann–Whitney U test. Categorical data were analyzed using the chi-square test or Fisher’s exact test. We applied PSM to minimize confounding factors associated with skeletal mass and sarcopenia. The variables were age, sex, mALBI grade, presence of sarcopenia, and BCAA administration. The model’s reliability was identified with the Hosmer–Lemeshow test. Spearman’s correlation coefficient was used to determine relationships between the variables. Time to lenvatinib treatment failure was calculated using the Kaplan–Meier method. All tests were two-sided, and *p* values < 0.05 were considered statistically significant. All statistical analyses were performed using SPSS Statistics for Windows, version 27 (IBM Corp., Armonk, NY, USA).

## 3. Results

### 3.1. Patient Characteristics

Baseline characteristics of patients in the two groups after PSM are summarized in Table 1. In total, 17 patients were included in the group without levocarnitine supplementation, and 17 patients were included in the group with levocarnitine supplementation. There were no significant differences in baseline factors between the two groups except for des-γ-carboxy prothrombin levels. Sarcopenia was present in 76% (28/34) of the patients at baseline.

### 3.2. Changes in SMI

Time courses of SMI values for patients without and with levocarnitine supplementation after the initiation of lenvatinib therapy are illustrated in Figure 2A. Among cases without levocarnitine supplementation, SMI values at 6 and 12 weeks after starting treatment were significantly lower compared with baseline (*p* = 0.035 and *p* = 0.030, respectively). In contrast, there were no significant differences in SMI at 6 or 12 weeks after starting treatment in patients with levocarnitine supplementation (*p* = 0.093 and *p* = 0.156, respectively). The changes in SMI from baseline (ΔSMI) at 6 and 12 weeks in the two groups are shown in Figure 2B. ΔSMI values at 6 and 12 weeks were significantly higher in the levocarnitine supplementation group compared with in the non-supplementation group (2.35 (−1.415–9.95] cm^2^/m^2^ vs. −2.63 (−10.885–0.1800] cm^2^/m^2^, *p* = 0.009; 2.000 (−2.070–6.690] cm^2^/m^2^ vs. −4.89 (−8.5425–2.4875] cm^2^/m^2^, *p* = 0.018, respectively).

### 3.3. Dynamic Changes in Carnitine Concentrations

Time courses of serum carnitine concentrations in patients without and with levocarnitine supplementation are presented in Figure 3. The analysis of cases without levocarnitine supplementation showed no significant differences in serum free carnitine levels between baseline and at 2, 4, and 6 weeks (*p* = 0.758, *p* = 0.586, and *p* = 0.193, respectively); however, serum free carnitine levels were significantly higher compared with baseline at 12 weeks (*p* = 0.038). In contrast, cases with levocarnitine supplementation showed significantly higher serum free carnitine levels at 2, 4, 6, and 12 weeks compared with baseline (all *p* < 0.001).

Serum acyl-carnitine levels were significantly higher compared with baseline at 2, 4, 6, and 12 weeks in both patients without (*p* = 0.013, *p* = 0.012, *p* = 0.002, and *p* = 0.003, respectively) and with levocarnitine supplementation (all *p* < 0.001). The acyl-to-free carnitine ratios were significantly higher compared with baseline at 2, 4, 6, and 12 weeks in both patients without (*p* = 0.011, *p* = 0.002, *p* = 0.003, and *p* = 0.003, respectively) and with levocarnitine supplementation (*p* = 0.028, *p* = 0.017, *p* = 0.019, and *p* = 0.001, respectively).

Although there were no significant differences in either serum-free carnitine or acyl-carnitine at baseline between the groups without and with levocarnitine supplementation (*p* = 0.0986 and *p* = 0.255, respectively), both serum free carnitine and acyl-carnitine levels were significantly higher at 2, 4, 6, and 12 weeks in the levocarnitine supplementation group than in the non-supplementation group (serum free carnitine: *p* < 0.001, *p* < 0.001, *p* < 0.001, and *p* = 0.006, respectively; acyl-carnitine: *p* = 0.012, *p* = 0.006, *p* = 0.006, and *p* = 0.016, respectively). 

In contrast, there were no significant differences in the acyl-to-free carnitine ratio at baseline or at 2, 4, 6 and 12 weeks between the two groups (*p* = 0.209, *p* = 0.904, *p* = 0.480, *p* = 0.770, and *p* = 0.468, respectively).

### 3.4. Changes in Serum Ammonia Levels

The changes in serum ammonia levels after beginning lenvatinib are shown in Figure 4. There were no significant differences in ammonia levels at baseline or at 2, 4, 6, and 12 weeks between the groups without and with levocarnitine supplementation (*p* = 0.850, *p* = 0.334, *p* = 0.639, *p* = 0.865, and *p* = 0.317, respectively).

### 3.5. Correlation between SMI and Carnitine Concentrations

Correlations between baseline SMI and at 6 and 12 weeks after the initiation of lenvatinib therapy with carnitine concentrations at the same time points are illustrated in Figure 5. Baseline SMI was significantly correlated with baseline serum free carnitine levels (r = 0.359, *p* = 0.037) (Figure 5A). Additionally, ΔSMI at 6 and 12 weeks were significantly positively correlated with the changes in serum free carnitine (r = 0.345, *p* = 0.045; r = 0.482, *p* = 0.008, respectively) (Figure 5D,G). Moreover, ΔSMI at 12 weeks was significantly positively correlated with the change in serum acyl-carnitine (r = 0.385, *p* = 0.039) (Figure 5H). Furthermore, there was a positive correlation between all SMI scores (at baseline and at 6 and 12 weeks) and all serum free carnitine values (r = 0.275, *p* = 0.006) (Figure 5J).

### 3.6. Comparison of Time to Treatment Failure in the Two Groups

Figure 6 shows the time to treatment failure, which was defined as the duration from the start of lenvatinib therapy to permanent discontinuation, for patients without and with levocarnitine supplementation. The median times to treatment failure for patients without and with levocarnitine supplementation were 142 days (95% confidence interval: 126–218 days) and 225 days (95% confidence interval: 189–283 days), respectively. Patients with levocarnitine supplementation tended to have a longer median time to treatment failure compared with patients without levocarnitine supplementation (*p* = 0.091).

## 4. Discussion

In this study, we analyzed the effect of levocarnitine supplementation in patients with HCC undergoing lenvatinib treatment, with a focus on changes in skeletal muscle mass and serum carnitine levels using a propensity score analysis. We found that sarcopenia progression in HCC patients during lenvatinib therapy was suppressed by levocarnitine supplementation. Additionally, our serum carnitine analysis revealed that serum-free carnitine levels were elevated in patients receiving levocarnitine supplementation during lenvatinib therapy.

We identified that patients without levocarnitine supplementation had a significant decrease in skeletal muscle mass during lenvatinib therapy. These results support the hypothesis described of Uchikawa et al. [13] that patients receiving lenvatinib experience a significant loss of skeletal muscle mass. Skeletal muscle mass is generally maintained by a balance between muscle protein synthesis and protein degradation. Previous studies have shown that inhibition of the phosphoinositide 3-kinase (PI3K)/thymoma viral proto-oncogene (AKT)/mechanistic target of rapamycin kinase (mTOR) pathway, which is an important intracellular signaling cascade for protein synthesis that is associated with skeletal muscle loss [22,23]. Additionally, tyrosine kinase receptors can activate the PI3K/AKT/mTOR pathway [23]. By inhibiting receptor tyrosine kinase signaling, TKIs including lenvatinib could indirectly suppress PI3K/AKT/mTOR signaling [7]. The median ΔSMI in patients without levocarnitine supplementation were −2.63 cm^2^/m^2^ and −4.89 cm^2^/m^2^ at 6 and 12 weeks in this study, respectively, demonstrating the need for an intervention strategy for skeletal muscle loss during TKI therapy.

We showed a significant inhibitory effect on the loss of skeletal muscle in patients with levocarnitine supplementation compared with those without levocarnitine supplementation during lenvatinib therapy. Our findings might be explained by a previous animal and human studies that showed that carnitine supplementation led to activation of the IGF-1/PI3K/AKT signaling pathway [24,25,26]. To the best of our knowledge, this is the first report to indicate that levocarnitine may play an important role in preventing muscle wasting during lenvatinib therapy.

We analyzed sarcopenia in relation to changes in serum carnitine. Recent studies have shown that TKI therapies for HCC such as sorafenib and lenvatinib were associated with drug-induced carnitine deficiency, probably due to the inhibitory effect of TKIs on the carnitine transporter [19,27]. Because serum acyl-carnitine levels were elevated 2 weeks, or later, after starting lenvatinib therapy in our study, the acyl-to-free carnitine ratio, which serves as an indicator of carnitine deficiency [28], was increased. These results suggest that lenvatinib-induced carnitine deficiency also occurred in our cohort. Our analysis of carnitine concentrations clearly showed that serum free carnitine levels had significantly increased in the group receiving levocarnitine supplementation at 2 weeks or later after starting lenvatinib therapy. Of note, levocarnitine supplementation led to increased serum acyl-carnitine levels during lenvatinib treatment. Thus, the acyl-to-free carnitine ratio was not improved in the group with levocarnitine supplementation. One possible explanation for this unexpected fluctuation is that serum acyl-carnitine is measured by an enzyme cycling method that involved the serum acetyl-carnitine level. Because carnitine also plays a role in eliminating acetyl-carnitine, which is an acetylated form of carnitine [29], elevations in serum acyl-carnitine levels may not necessarily reflect mitochondrial carnitine deficiency.

Hyperammonemia is a commonly observed in patients with liver cirrhosis and is considered a major driver of muscle atrophy [30,31]. In addition, a recent study suggested that the gut–liver–muscle axis is associated with sarcopenia in cirrhotic patients [32]. Previous studies have showed that levocarnitine supplementation reduces skeletal muscle mass loss by improving hyperammonemia, but also reduces the bone mineral density loss and frequency of muscle cramping in patients with chronic liver disease [33,34,35]. However, these studies have investigated the long-term effect of carnitine. Although recent studies indicate that an early elevation of serum ammonia occurred after the initiation of lenvatinib therapy [36,37], our data—from a relatively larger cohort—compared with the former studies showed no significant increase in serum ammonia after the initiation of lenvatinib therapy. Moreover, in our propensity-score-matched cohort, approximately 44% of patients classified as mALBI grade 1, including normal liver impairment. The interstudy discrepancy regarding the presence of significant serum ammonia elevation might be due to differences in cohort size and patient characteristics. Thus, research should focus on factors other than hyperammonemia leading to skeletal muscle loss during lenvatinib therapy. Because levocarnitine has the potential to counteract mitochondrial dysfunction [38], it is reasonable to suppose that the ameliorating effect of levocarnitine on mitochondrial function was also one of the factors associated with its inhibitory effect on lenvatinib-induced sarcopenia.

This study suggests that supplementation with 1500 mg/day of levocarnitine is a promising strategy to suppress lenvatinib-induced sarcopenia. Recent studies have found that the suppression of sarcopenia progression by levocarnitine supplementation in cirrhotic patients was dose-dependent and that high-dose levocarnitine supplementation (≥1274 mg/day) was associated with a reduction in serum ammonia levels [33]. Although we could not assess differences between different levocarnitine doses during lenvatinib therapy, 1500 mg/day of levocarnitine might be a relatively reasonable dose. A future observational study using various levocarnitine doses is needed, because it is important to determine the optimal dose of levocarnitine for preventing skeletal muscle loss during lenvatinib therapy. Taking into consideration medical economics, it will also be important to consider the type of patients who are recommended to receive carnitine supplementation at the initiation of lenvatinib therapy. Because BCAA supplementation has been reported to prevent the occurrence of sarcopenia in patients with liver cirrhosis [39], we selected prior BCAA treatment for PSM. Further clinical research is also needed with regard to the impact of supplementation with a combination of levocarnitine and BCAA. Furthermore, as levocarnitine is metabolized into trimethylamine N-oxide, recent studies have showed that long-term levocarnitine supplementation may elevate circulating trimethylamine N-oxide [40], which is associated with a risk of cardiovascular disease [41]. Therefore, clinicians must carefully consider the potential long-term effects of levocarnitine on the cardiovascular system.

The relationship between serum carnitine levels and skeletal muscle volume has not been identified. The most recent study has shown that changes in serum-free carnitine levels after hepatitis C virus eradication by directly acting antivirals are associated with increased skeletal muscle mass [42]. These results are in line with our findings that baseline free carnitine levels and changes in those levels correlated with skeletal muscle volume. Additionally, we found that serum free carnitine levels at any timepoint during lenvatinib therapy were associated with skeletal muscle volume at the same timepoint. As mentioned above, serum free carnitine might be more or less related to skeletal muscle volume. Increased circulating free carnitine levels in patients with levocarnitine supplementation may reflect the amelioration of carnitine deficiency in skeletal muscle.

Decreased skeletal mass is associated with the occurrence of severe adverse events and decreased time to treatment failure and overall survival [12]. In our study, we found that patients with carnitine supplementation tended to have a prolonged median time to treatment failure compared with patients without supplementation. However, controlled trials are needed to obtain definitive evidence of the clinical efficacy of levocarnitine supplementation such as time to treatment failure, progression-free survival, post-progression survival, and overall survival.

Our study had some limitations. First, this study had a single-institution retrospective design and included a comparatively limited number of patients. Nevertheless, it should be emphasized that the timing of CT examinations was fixed, and dynamic changes in the serum carnitine concentrations could be precisely analyzed because of the single-institution nature of the study. Second, because PSM was conducted in this study, the number of enrolled patients was limited. However, a strength of this study was that its statistical reliability was improved by PSM. Third, we did not examine any markers such as insulin-like growth factor 1 or myostatin, which are associated with skeletal muscle conditions [43]. Despite these limitations, our findings including the serum carnitine analysis provide evidence that levocarnitine supplementation can contribute to preventing skeletal muscle depletion during lenvatinib therapy in HCC patients. 

## 5. Conclusions

In conclusion, our findings suggest that lenvatinib therapy for HCC is a high-risk procedure for progression and/or development of sarcopenia but that levocarnitine supplementation can suppress sarcopenia progression during lenvatinib therapy. 

## Figures and Tables

**Figure 1 nutrients-13-04428-f001:**
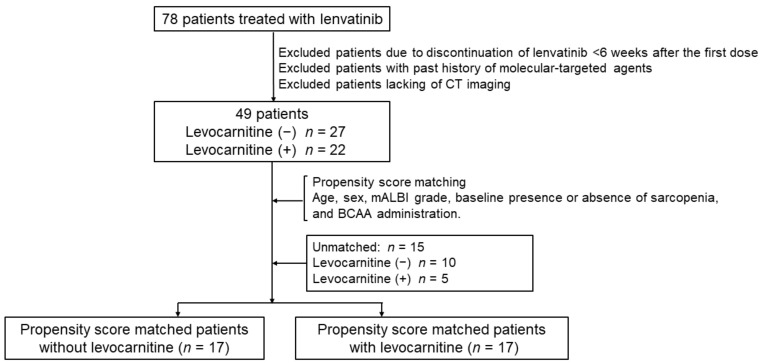
Flow chart of patients included in this study. BCAA—branched-chain amino acid; CT—computed tomography; mALBI—modified albumin-bilirubin grade.

**Figure 2 nutrients-13-04428-f002:**
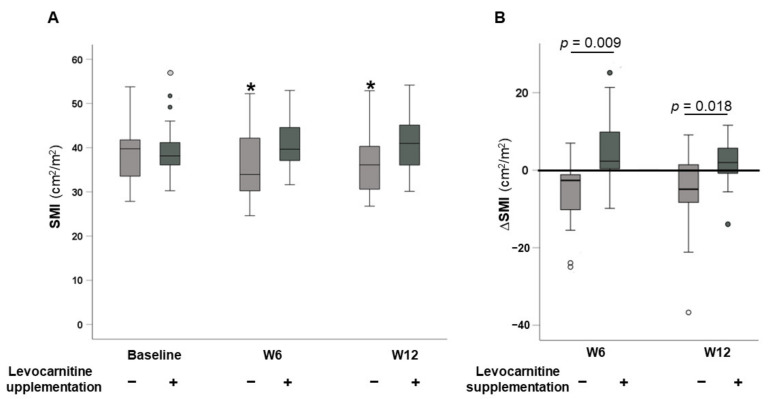
(**A**) Time course of skeletal muscle index (SMI) values and (**B**) changes in SMI from baseline (ΔSMI) between patients without and with levocarnitine supplementation at 6 and 12 weeks after initiating lenvatinib therapy. * *p* < 0.05 vs. baseline.

**Figure 3 nutrients-13-04428-f003:**
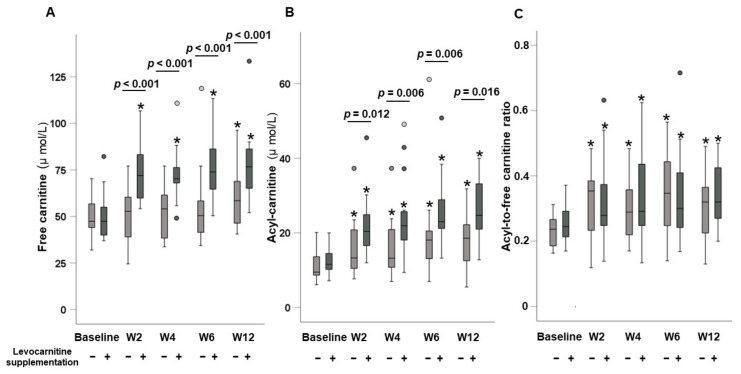
Time course of (**A**) serum free carnitine levels, (**B**) serum acyl-carnitine, and (**C**) the acyl-to-free carnitine ratio between patients without and with levocarnitine supplementation after lenvatinib therapy. * *p* < 0.05 vs. baseline.

**Figure 4 nutrients-13-04428-f004:**
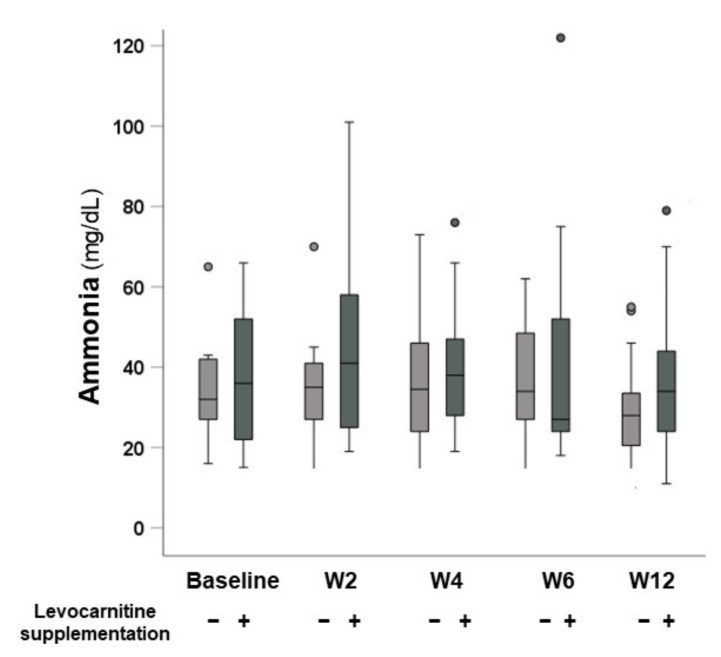
Time courses of serum ammonia levels in patients without and with levocarnitine supplementation.

**Figure 5 nutrients-13-04428-f005:**
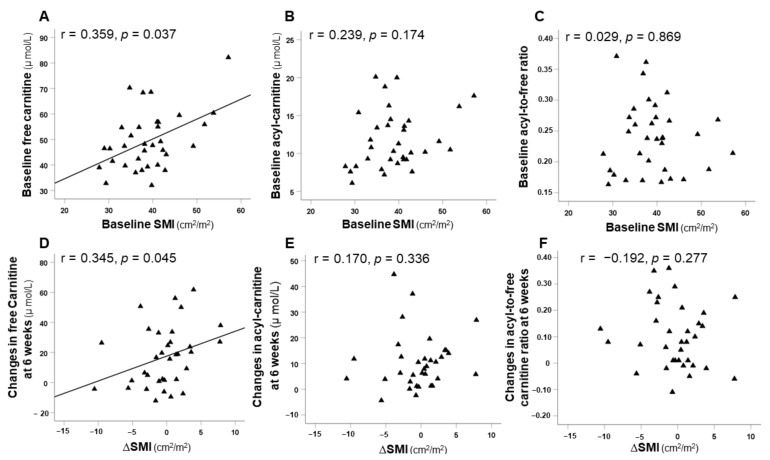
Correlations between baseline skeletal muscle index (SMI) scores and serum carnitine levels (**A**–**C**), correlations between changes in SMI from baseline (ΔSMI) and changes in serum carnitine levels (**D**–**I**) at 6 and 12 weeks, and correlation between all SMI scores and all free carnitine values over 12-week period (**J**).

**Figure 6 nutrients-13-04428-f006:**
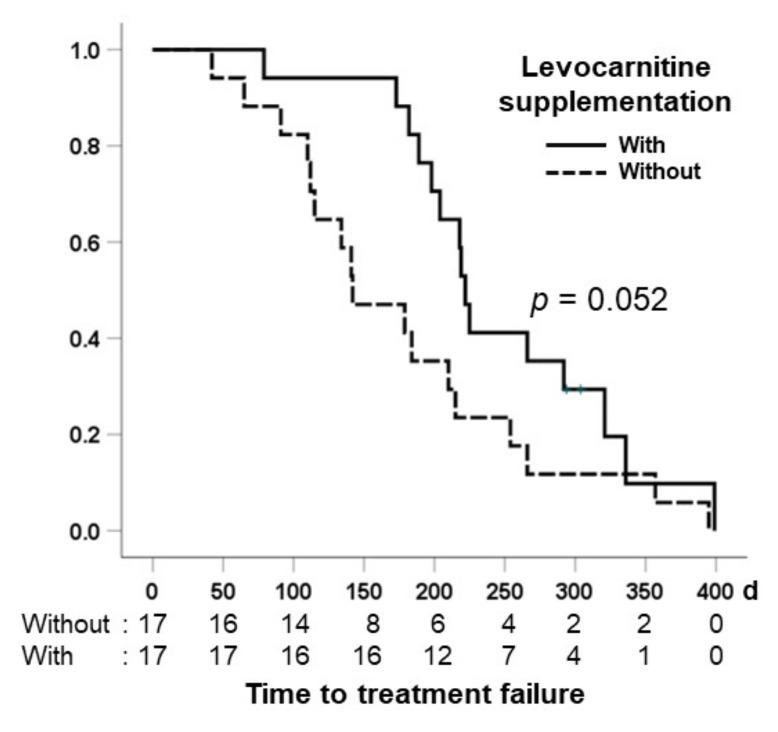
Comparison of the time to treatment failure between patients without and with levocarnitine supplementation.

**Table 1 nutrients-13-04428-t001:** Comparison of baseline characteristics in patients without and with levocarnitine supplementation.

	All Patients (n = 34)	Without Levocarnitine Supplementation (n = 17)	With Levocarnitine Supplementation n = 17	*p* Value
Age, years	77.5 (52–90)	77 (67–90)	79 (52–87)	0.959
Sex, male/female	28/6	15/2	13/4	0.656
HBV/HCV/NBNC	4/10/20	2/5/10	2/5/10	1.000
ECOG PS, 0/1	29/5	14/3	15/2	1.000
Naïve, yes/no	30/2	15/1	15/1	1.000
Body weight, kg	59.1 (38.6–81.4)	59.0 (38.6–81.4)	61 (41.2–69.1)	0.959
BMI, kg/m^2^	22.82 (15.88–27.77)	21.82 (17.25–27.77)	23.31 (15.58–27.62)	0.221
Initial dose, 12 mg/8 mg	15/19	7/10	8/9	0.730
BCLC staging, B/C	23/11	11/6	12/5	0.714
Albumin, g/dL	3.8 (2.9–4.7)	3.7 (3.3–4.2)	3.9 (2.9–4.7)	0.407
Total bilirubin, mg/dL	0.7 (0.3–1.4)	0.6 (0.4–1.2)	0.9 (0.4–1.4)	0.061
Platelet count, ×10^4^/μL	18.2 (7.6–46.7)	20.4 (7.6–35.5)	16.0 (7.8–46.7)	0.865
Ammonia, mg/dL	33.5 (15–66)	32 (16–65)	36 (15–66)	0.850
Free carnitine, µmol/L	47.4 (32–82.2)	47.4 (32–70.3)	47.4 (37–82.2)	0.986
Acyl-carnitine, µmol/L	10.65 (6.1–20.1)	9.5 (6.1–20.1)	11.6 (7.2–20.0)	0.255
AC/FC ratio	0.24 (0.16–0.37)	0.24 (0.16–0.31)	0.25 (0.17–0.37)	0.221
AFP, ng/mL	32.1 (0.9–70,000)	21.2 (0.9–35,500)	85.5 (1.6–70,000)	0.730
DCP, mAU/mL	215.5 (16–5,520,000)	1730 (21–72,300)	63 (15–10,000)	0.008
Child–Pugh score, 5/6	24/10	11/6	13/4	0.452
ALBI score	−2.51 (−3.21–−1.64)	−2.56 (−3.02–−2.11)	−2.51 (−3.11–−1.64)	0.783
mALBI	15/12/7	7/7/3	8/5/4	0.543
BCAA supplementation, yes/no	11/23	5/12	6/11	0.714
SMI, cm^2^/m^2^	38.45 (29.01–59.79)	39.75 (27.86–53.77)	38.15 (30.27–57.14)	0.796
Sarcopenia, yes/no	26/8	13/4	13/4	1.000

Median (range) or n; AC/FC—acyl-to-free carnitine; AFP—α-fetoprotein; BCLC—Barcelona Clinic Liver Cancer; BCAA—branched-chain amino acid; BMI—body mass index; DCP—des-γ-carboxy prothrombin; ECOG PS—Eastern Cooperative Oncology Group performance status; HBV—hepatitis B virus; HCV—hepatitis C virus; mALBI—modified albumin–bilirubin; SMI—skeletal muscle index.

## Data Availability

The datasets that support the findings of this study are available from the corresponding author upon reasonable request.

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
