# Peer review of "Levocarnitine Supplementation Suppresses Lenvatinib-Related Sarcopenia in Hepatocellular Carcinoma Patients: Results of a Propensity Score Analysis"

_nutrients, 2021, doi:10.3390/nu13124428_

Round 1

Reviewer 1 Report

Major concerns

Introduction

- lines 34-38: the information is not relevant; the abbreviations VEGFR, FGFR, PDGFR are not used in any other part of the manuscript;

- lines 48-49: make sure that the study of Uchikawa et al. [13] is well described;

Methods

- lines 112-117: specify where the carnitine was determined – plasma or serum; add centrifugation details and information about methods of other markers measurements (at least ammonia)

Results

- SMI and circulating carnitine results are presented for baseline, 6W, and 12W, but the correlations only for baseline and delta at 6W. The results of delta 12W should also be included. Moreover, the correlation of all values (all subjects, 3 times = 102 points) SMI and free carnitine could give better picture, whether there is any relation between SMI and circulating free carnitine;

- Table 1: specify the meaning ‘Performance status’ – describe in the legend or in methods section;

- lines 150-151: make sure the values are correct (2.35 [-1.415 – 1.95]), the median value is out of range values;

- Figures 2,3,4: specify the meaning of the grey asterisks, and the numbers next to outliers; or prepare more clear figures;

Discussion

- lines 234-237: the results have been presented in the results section, omit the values in the discussion section

- lines 238-244: ref. 24 represents results of the animal model study; discuss also the results of the human studies (e.g. PMIDs: 28115977, 29473908).

- lines 248-250: add the reference to the information: ‘the acyl-to-free carnitine ratio, serves as an indicator of carnitine deficiency’

- lines 258-264: consider a separate paragraph on ‘mitochondrial carnitine deficiency’

- lines 262-264: ref. 27 does not present results of skeletal muscle mass loss prevention; discuss obtained results comparing to previously reported: skeletal muscle mass index (PMID: 32973943), and skeletal muscle volume (PMID: 31150367); consider carnitine dose as a potential factor (ref. 28);

- lines 264-271: confront ammonia results with previously reported (e.g. PMIDs: 31431642, 32878816);

- lines 287-294: the results of all values correlation (all subjects, 3 times = 102 points) can improve this part of discussion;

- lines 296-298: explain to the readers in a simplified way ‘time to treatment failure’

- lines 313-315: despite the fact that L-carnitine may regulate skeletal muscle protein balance, the supplementation may elevate circulating trimethylamine N-oxide (e.g. PMID: 32958033); the note about TMAO, should be added as an information about potential risk of L-carnitine supplementation;

Minor

- Figure 1: ‘Unmatched’ box – spelling ‘Levoaarnitine’

- line 410: correct the references numbering

Author Response

Reviewer #1: Comments and Suggestions for Authors

Major concerns

Introduction

- lines 34-38: the information is not relevant; the abbreviations VEGFR, FGFR, PDGFR are not used in any other part of the manuscript;

Response: We have deleted the abbreviations VEGFR, FGFR, and PDGFR.

- lines 48-49: make sure that the study of Uchikawa et al. [13] is well described;

Response: In accordance with your helpful comment, we have changed ‘Uchikawa et al.’ to ‘A recent study’.

- lines 112-117: specify where the carnitine was determined – plasma or serum; add centrifugation details and information about methods of other markers measurements (at least ammonia)

Response: Thank you for this comment. We have specified where the serum carnitine was determined and have added centrifugation details in the revised manuscript. In addition, we have changed ‘plasma’ to ‘serum’ throughout the revised manuscript. Because we added the methods used to obtain the serum ammonia measurements, we changed the subsection title to ‘Measuring serum ammonia and carnitine content’ and included this information on lines 112–119 of the revised manuscript.

Results

- SMI and circulating carnitine results are presented for baseline, 6W, and 12W, but the correlations only for baseline and delta at 6W. The results of delta 12W should also be included. Moreover, the correlation of all values (all subjects, 3 times = 102 points) SMI and free carnitine could give better picture, whether there is any relation between SMI and circulating free carnitine;

Response: Thank you for this insightful comment. We analyzed the correlation between delta SMI and circulating carnitine at 12 weeks. In addition, we analyzed the relationships between the SMI scores and all serum free carnitine values. The results have been added to Figure 5 in the revised manuscript. In addition, an explanation of the results shown in Figure 5 has been added on lines 201–205 of the revised manuscript.

- Table 1: specify the meaning ‘Performance status’ – describe in the legend or in methods section;

Response: Thank you for this comment. We have replaced the term ‘Performance status’ with ‘ECOG PS’ in Table 1 and added ‘ECOG PS, Eastern Cooperative Oncology Group performance status’ in the footnote.

- lines 150-151: make sure the values are correct (2.35 [-1.415 – 1.95]), the median value is out of range values;

Response: Thank you for pointing out this error. The error has been corrected on line 155 of the revised manuscript.

- Figures 2,3,4: specify the meaning of, and the numbers next to outliers; or prepare more clear figures;

Response: Thank you for this comment about improving the quality of original Figure 2, 3 and 4. Because the grey asterisks also indicate outliers, we have changed the grey asterisks into grey circles and have deleted the numbers next to the outliers in Figures 2, 3, and 4 of the revised manuscript.

Discussion

- lines 234-237: the results have been presented in the results section, omit the values in the discussion section

Response: In accordance with this comment, we have removed the results from the Discussion section.

- lines 238-244: ref. 24 represents results of the animal model study; discuss also the results of the human studies (e.g. PMIDs: 28115977, 29473908).

Response: Thank you for your helpful comment. We have also cited human studies (PMIDs 28115977 and 29473908) on lines 244–246 of the revised manuscript.

- lines 248-250: add the reference to the information: ‘the acyl-to-free carnitine ratio, serves as an indicator of carnitine deficiency’

Response: In accordance with this comment, we have added a reference on line 254 of the revised manuscript.

- lines 258-264: consider a separate paragraph on ‘mitochondrial carnitine deficiency’

Response: Thank you for this comment. We intended to state the reason for serum acyl-carnitine elevation in the group with levocarnitine supplementation during lenvatinib treatment in terms of acyl-carnitine behavior. In addition, we have deleted the phrase ‘mitochondrial function’ on line 264 of the original manuscript in response to the next comment from the reviewer. Therefore, to maintain consistency throughout the manuscript, we have not created a separate paragraph on ‘mitochondrial carnitine deficiency’.

- lines 262-264: ref. 27 does not present results of skeletal muscle mass loss prevention; discuss obtained results comparing to previously reported: skeletal muscle mass index (PMID: 32973943), and skeletal muscle volume (PMID: 31150367); consider carnitine dose as potential factor (ref. 28);

Response: Thank you for your helpful comment. We have deleted ref. 27 from this sentence in the revised manuscript. We added further discussion and cited PMIDs 32973943 and 31150367 on line 268–273 of the revised manuscript. Although the reviewer would like us to discuss the carnitine dose, we have discussed this point in the next paragraph of the original manuscript (lines 286–295 of the revised manuscript).

- lines 264-271: confront ammonia results with previously reported (e.g. PMIDs: 31431642, 32878816);

Response: Thank you for this important comment. As the reviewer pointed out, the sentence may be too strong. Therefore, we discussed the reasons for the lack of a relationship between the serum ammonia level and sarcopenia progression on lines 273–279 of the revised manuscript. Moreover, we have toned down the language in the revised manuscript.

- lines 287-294: the results of all values correlation (all subjects, 3 times = 102 points) can improve this part of discussion;

Response: Thank you for this comment. Because the serum free carnitine levels at any timepoint during lenvatinib therapy were associated with skeletal muscle volume at the same timepoint, we have added the content to strengthen our hypothesis that serum free carnitine might be related to skeletal muscle volume on lines 310–311 of the revised manuscript.

- lines 296-298: explain to the readers in a simplified way ‘time to treatment failure’

Response: In accordance with this comment, we have added the definition of ‘time to treatment failure’ on lines 211–212 of the revised manuscript.

- lines 313-315: despite the fact that L-carnitine may regulate skeletal muscle protein balance, the supplementation may elevate circulating trimethylamine N-oxide (e.g. PMID: 32958033); the note about TMAO, should be added as an information about potential risk of L-carnitine supplementation;

Response: Thank you for this clinically important comment. We have added the potential risks of long-term levocarnitine supplementation on lines 300–305 of the revised manuscript.

- Figure 1: ‘Unmatched’ box – spelling ‘Levoaarnitine’

Response: The error has been corrected.

- line 410: correct the references numbering

Response: The errors in the reference list have been corrected.

Reviewer 2 Report

Dear Authors,

In this paper you aimed at investigating the inhibitory effect of levocarnitine supplementation on sarcopenia development/progression in HCC patients treated with lenvatinib, an oral tyrosine kinase inhibitor that can exacerbate muscle failure.

The topic is very interesting, and the manuscript is well written.

On the other hand, I suggest some minor revisions to improve this work.

Minor revisions

INTRODUCTION AND DISCUSSION SECTIONS.

According to EWGSOP, sarcopenia is a progressive and generalized skeletal muscle disorder that is associated with increased likelihood of adverse outcomes including falls, fractures, physical disability and mortality. Sarcopenia is largely attributable to ageing; however, in many cases, other causes can be identified, such as malignancy, inflammatory bowel diseases, malnutrition, physical inactivity, or organ failure. In this context, the emerging concept of the “gut-liver-muscle axis” should be adequately described in the pathogenesis of HCC-related sarcopenia, taking into the key role of inflammation and gut microbiota in the development of muscle failure.

Thus, please improve these sections, citing more studies present in literature.

Additional reference recommended:

- Ponziani FR, et al. Characterization of the gut-liver-muscle axis in cirrhotic patients with sarcopenia. Liver Int. 2021 Jun;41(6):1320-1334. doi: 10.1111/liv.14876.

- Nardone OM, et al. Inflammatory Bowel Diseases and Sarcopenia: The Role of Inflammation and Gut Microbiota in the Development of Muscle Failure. Front Immunol. 2021 Jul 13;12:694217. doi: 10.3389/fimmu.2021.694217.

- Rinninella E, et al. Skeletal Muscle Loss during Multikinase Inhibitors Therapy: Molecular Pathways, Clinical Implications, and Nutritional Challenges. Nutrients. 2020 Oct 12;12(10):3101. doi: 10.3390/nu12103101.

Author Response

Response to the Reviewer’ Comments

Reviewer #2: In this paper you aimed at investigating the inhibitory effect of levocarnitine supplementation on sarcopenia development/progression in HCC patients treated with lenvatinib, an oral tyrosine kinase inhibitor that can exacerbate muscle failure.

The topic is very interesting, and the manuscript is well written.On the other hand, I suggest some minor revisions to improve this work.

Minor revisions

INTRODUCTION AND DISCUSSION SECTIONS.

-According to EWGSOP, sarcopenia is a progressive and generalized skeletal muscle disorder that is associated with increased likelihood of adverse outcomes including falls, fractures, physical disability and mortality. Sarcopenia is largely attributable to ageing; however, in many cases, other causes can be identified, such as malignancy, inflammatory bowel diseases, malnutrition, physical inactivity, or organ failure. In this context, the emerging concept of the “gut-liver-muscle axis” should be adequately described in the pathogenesis of HCC-related sarcopenia, taking into the key role of inflammation and gut microbiota in the development of muscle failure.Thus, please improve these sections, citing more studies present in literature.

Thus, please improve these sections, citing more studies present in literature.

Additional reference recommended:

- Ponziani FR, et al. Characterization of the gut-liver-muscle axis in cirrhotic patients with sarcopenia. Liver Int. 2021 Jun;41(6):1320-1334. doi: 10.1111/liv.14876.

- Nardone OM, et al. Inflammatory Bowel Diseases and Sarcopenia: The Role of Inflammation and Gut Microbiota in the Development of Muscle Failure. Front Immunol. 2021 Jul 13;12:694217. doi: 10.3389/fimmu.2021.694217.

- Rinninella E, et al. Skeletal Muscle Loss during Multikinase Inhibitors Therapy: Molecular Pathways, Clinical Implications, and Nutritional Challenges. Nutrients. 2020 Oct 12;12(10):3101. doi: 10.3390/nu12103101.

Response: We thank the reviewer for this comment. In accordance with this comment, we have cited and added the recommended reference of Ponziani FR et al. (PMID: 33713524) on lines 267–268 of the revised manuscript. The other reference that the reviewer recommended (Skeletal Muscle Loss during Multikinase Inhibitors Therapy by Rinninella et al., PMID: 33053632) has already been cited as ref. 7 in the Introduction and Discussion sections of the original manuscript.
